# Prevalence and Molecular Characterization of *Cryptosporidium* spp., *Giardia duodenalis*, and *Enterocytozoon bieneusi* in Cattle in Heilongjiang Province, Northeast China

**DOI:** 10.3390/ani14111635

**Published:** 2024-05-30

**Authors:** Jun-Feng Gao, Lu Zhou, Ai-Hui Zhang, Mei-Ru Hou, Xue-Wei Liu, Xin-Hui Zhang, Jia-Wen Wang, Xue Wang, Xue Bai, Chen-Long Jiao, Yan Yang, Zhuo Lan, Hong-Yu Qiu, Chun-Ren Wang

**Affiliations:** Key Laboratory of Bovine Disease Control in Northeast China, Ministry of Agriculture and Rural Affair, Key Laboratory of Prevention and Control of Zoonotic Diseases of Daqing, College of Animal Science and Veterinary Medicine, Heilongjiang Bayi Agricultural University, Daqing 163319, China; gaojunfeng_2005@163.com (J.-F.G.); zhouluu115@163.com (L.Z.); zhangaihui2090@163.com (A.-H.Z.); houmeiru168@126.com (M.-R.H.); 19845919553@163.com (X.-W.L.); zzzhangxinhui@163.com (X.-H.Z.); aa18345481299@163.com (J.-W.W.); wangxue20210427@163.com (X.W.); baixueee77@163.com (X.B.); jiaochenlong888@163.com (C.-L.J.); m15685258867@163.com (Y.Y.); lannzhuo@163.com (Z.L.); qiuhongyu95@163.com (H.-Y.Q.)

**Keywords:** *Cryptosporidium* spp., *Giardia duodenalis*, *Enterocytozoon bieneusi*, prevalence, cattle, Heilongjiang province

## Abstract

**Simple Summary:**

*Cryptosporidium* spp., *Giardia duodenalis*, and *Enterocytozoon bieneusi* are among the most common diarrhea-causing pathogens in humans and livestock. This was the first study to systematically and molecularly detect the prevalence of three intestinal protozoa in cattle in Heilongjiang Province, Northeast China. Our findings indicate that a variety of species/genotypes are prevalent in cattle in Heilongjiang Province and that some of them are zoonotic, indicating a risk of zoonotic disease transmission in endemic areas. This study helps to understand the epidemiology of *Cryptosporidium* spp., *Giardia duodenalis*, and *Enterocytozoon bieneusi* in cattle in Heilongjiang Province.

**Abstract:**

*Crytosporidium* spp., *Giardia duodenalis*, and *Enterocytozoon bieneusi* are important diarrheal pathogens with a global distribution that threatens the health of humans and animals. Despite cattle being potential transmission hosts of these protozoans, the associated risks to public health have been neglected. In the present study, a total of 1155 cattle fecal samples were collected from 13 administrative regions of Heilongjiang Province. The prevalence of *Cryptosporidium* spp., *G. duodenalis*, and *E. bieneusi* were 5.5% (64/1155; 95% CI: 4.2–6.9), 3.8% (44/1155; 95% CI: 2.7–4.9), and 6.5% (75/1155; 95% CI: 5.1–7.9), respectively. Among these positive fecal samples, five *Cryptosporidium* species (*C. andersoni*, *C. bovis*, *C. ryanae*, *C. parvum*, and *C. occultus*), two *G. duodenalis* assemblages (E and A), and eight *E. bieneusi* genotypes (BEB4, BEB6, BEB8, J, I, CHS7, CHS8, and COS-I) were identified. Phylogenetic analysis showed that all eight genotypes of *E. bieneusi* identified in the present study belonged to group 2. It is worth noting that some species/genotypes of these intestinal protozoans are zoonotic, suggesting a risk of zoonotic disease transmission in endemic areas. The findings expanded our understanding of the genetic composition and zoonotic potential of *Cryptosporidium* spp., *G. duodenalis*, and *E. bieneusi* in cattle in Heilongjiang Province.

## 1. Introduction

*Cryptosporidium* spp., *Giardia duodenalis*, and *Enterocytozoon bieneusi* are among the most common diarrhea-causing pathogens with a global distribution in humans and livestock. These intestinal protozoans can lead to severe chronic diarrhea, malnutrition, and sometimes even death in immunocompromised or deficient hosts [1]. The (oo)cysts/spores produced by these protozoans are widely distributed in the environment after they are excreted with the feces of hosts. Humans and livestock may then be infected via the fecal–oral route, causing cryptosporidiosis, giardiasis, or enterosporosis. Due to the lack of effective drugs and commercial vaccines for three protozoan diseases, this may result in a substantial economic impact from livestock losses [2,3,4].

*Cryptosporidium* spp. infection is characterized by diarrhea in neonatal livestock, the main symptoms of infected calves include severe diarrhea, dehydration, growth retardation, and sometimes death [4]. As cryptic carriers, adult livestock play a vital role in epidemiology, causing reinfection at the herd level, even when individual animals appear asymptomatic during the infection period [3]. To date, 44 recognized species have been identified in the genus *Cryptosporidium*, and at least 12 *Cryptosporidium* species have been reported in cattle worldwide [5]. Among these, the dominant cattle-infecting species are *C. andersoni*, *C. parvum*, *C. bovis*, and *C. ryanae* [6,7]. It has also been reported that *C. parvum*, *C. andersoni*, and *C. bovis* can infect humans, with zoonotic cryptosporidiosis being mainly caused by *C. parvum* [8]. In short, cryptosporidiosis represents a zoonotic risk with the potential to present a large-scale outbreak.

*Giardia* spp. are opportunistic intestinal pathogens found in numerous species of mammalian hosts [9]. The genus name *Giardia* was initially established by Kunstler (1882), and six valid species have been currently identified based on morphology, namely, *G. duodenalis*, *Giardia muris*, *Giardia microti*, *Giardia psittaci*, *Giardia ardeae*, and *Giardia agilis*. Among these, *G. duodenalis* is the only species that causes human infection [5]. *G. duodenalis* has been subclassified into eight genetic assemblages (A–H). Of these, assemblages A and B are characterized by a wide host range, being capable of infecting most mammals, including humans; indeed, A and B are the only assemblages reported to cause human infections. Assemblages C and D mainly infect canines. Assemblage E mainly affects artiodactyl hosts such as cattle, sheep, swine, horses, and other livestock. Assemblage F mainly infects cats. Assemblage G mainly infects rodents. Assemblage H predominantly targets wild animals such as seals and marsupials [10]. Finally, assemblages E and A are the most common genotypes of *G. duodenalis* in cattle, which carry the risk of potentially infecting humans [11,12].

*Enterocytozoon bieneusi* is a fungus-like intestinal protozoan that can cause symptomatic and asymptomatic effects in a wide range of terrestrial animals worldwide [13]. *E. bieneusi* is characterized by high genetic diversity in hosts, and at least 600 genotypes have been identified thus far [14]. Phylogenetic assessments based on ITS rRNA gene fragments have shown that these genotypes are distributed in 11 major genetic groups. Genotypes in different groups exhibit varying degrees of host specificity and zoonotic potential. Group 1 is the largest group and includes the majority of genotypes isolated from almost all current known host types, including humans. The members of group 2 are the most common causes of *E. bieneusi* infection in ruminants such as cattle, deer, sheep, and goats. Group 2 is therefore considered to be adapted to ruminants; however, some of the genotypes in group 2 also exist in humans and are zoonotic. Groups 3–11 include largely host-adapted genotypes associated with specific animal species [13,15]. Though it has long been known that cattle are potential transmission hosts of *E. bieneusi*, the associated risks to public health have largely been neglected. It is also well known that the fecal–oral model is the primary transmission route for the three intestinal protozoan pathogens named above, though infection can also be spread by contaminated water, food, and fomites [16]. Notably, adult cattle, as negative carriers, continuously spread pathogens outward, and their manure is a major source of water and food contamination [17,18]. In light of this, more molecular-based epidemiological studies should be conducted to assess the risk of potential zoonotic events and to prevent and control the occurrence of these protozoan pathogens.

Heilongjiang Province is a major agricultural province, and the mainstay of beef and dairy cattle breeding, which is located on the northeastern of China. Despite cattle being the potential transmission hosts of these zoonotic intestinal protozoa, they continue to spread pathogens outward continuously, nevertheless, the associated risks to public health have largely been neglected. Thus, more molecular-based epidemiological studies should be needed to assess the risk of potential zoonotic events and to prevent and control these protozoan pathogens. Despite several studies that have reported about these protozoan separate reports in Heilongjiang Province, these data are now almost a decade old [19,20,21,22,23]. In the present study, we aim to conduct an up-to-date assessment of the prevalence, genetic characteristics, and public health potential of *Cryptosporidium*, *G. duodenalis,* and *E. bieneusi* in fecal samples taken from cattle in Heilongjiang Province.

## 2. Materials and Methods

### 2.1. Study Areas and Sample Collection

Between May 2022 and July 2023, 1155 fresh fecal specimens, including 570 from beef cattle and 585 from dairy cattle, were randomly collected via rectal sampling in 13 administrative regions of Heilongjiang Province in Northeast China (121°11′ W, 135°05′ E, 43°26′ S, 53°33′ N) (Figure 1). Fecal specimens were taken from 722 female cattle and 433 male cattle. A total of 273 fecal specimens were from cattle younger than 12 months, 421 were from cattle aged 12–18 months, and 461 were from cattle older than 18 months. Each specimen was individually packaged to prevent cross-contamination between samples. Specimen information was recorded during sampling; packaged specimens were then transported to the laboratory, stored at −20 °C, and detected within 48 h.

### 2.2. Genomic DNA Extraction and PCR Amplifications

Total genomic DNA was extracted from approximately 200 mg of each fecal specimen using a Genomic DNA Extraction Kit (Solarbio, Beijing, China), according to the manufacturer’s recommended protocol. All DNA samples were kept at −20 °C until needed for subsequent experiments.

The prevalence of *Cryptosporidium* spp. was detected via the nested PCR amplification of the small-subunit ribosomal RNA (*SSU rRNA*) gene (~830 bp) [24]. The prevalence of *G. duodenalis* was detected via the nested PCR amplification of the *beta-giardin* gene (*bg*) (~511 bp) [25]. The prevalence of *E. bieneusi* was detected via the nested PCR amplification of the internal transcribed spacer (ITS) gene (~392 bp) [26] (Table 1). PCR reactions were performed using a thermocycler (BioRad, Hercules, CA, USA) using 2 µL genomic DNA (the primary PCR) or 1 µL PCR amplification product (the secondary PCR), along with 2.5 µL 10 × Ex Taq Buffer (Takara, Dalian, China), 400 mM dNTP Mixture (Takara, Dalian, China), 0.2 µL TaKaRa Ex Taq polymerase (Takara, Dalian, China), 0.5 µM of each primer (Qingke Biology, Harbin, China), and the addition of ddH_2_O up to 25 µL.

The amplification conditions for primary and secondary PCR are consistent for *Cryptosporidium* spp., *G. duodenalis*, and *E. bieneusi*. The PCR reaction was performed under the following conditions: 95 °C for 2 min (initial denaturation), then 95 °C for 30 s (denaturation), 55 °C, 60 °C, or 57 °C for 30 s (annealing), 72 °C for 1 min (extension) for 30 cycles, and a final extension at 72 °C for 8 min. The secondary PCR products were detected in a 0.8% (*w*/*v*) agarose gel upon ethidium bromide staining.

### 2.3. Sequencing and Phylogenetic Analysis

All positive secondary PCR products were sent to Sangon Biotech Ltd. (Shanghai, China) for sequencing from both directions. The obtained sequences were assembled using DNAStar v5.0 software [27] and aligned with the sequences available in the NCBI nucleotide database (http://blast.ncbi.nlm.nih.gov/Blast, accessed on 20 November 2023). The genotypes or subtypes of *Cryptosporidium* spp, *G. duodenalis*, and *E. bieneusi* were determined by examining multiple sequences aligned using Clustal X v1.83 software (http://www.clustal.org/, accessed on 20 November 2023), and phylogenetic trees were constructed via the maximum likelihood (ML) method using MEGA v11.0 software under the best-fit substitution model [28]. All modeling was carried out under the assumption that the variation rate among these sites was gamma-distributed. Bootstrap evaluations were inferred from 1000 replicates. The phylograms were drawn using FigTree v1.4.4 software (http://tree.bio.ed.ac.uk/software/figtree/, accessed on 25 November 2023).

Representative nucleotide sequences obtained in the present study were available in GenBank with the accession numbers PP669781-PP669785 for *Cryptosporidium* spp., PP706152-PP706155 for *G. duodenalis*, and PP692496-PP692503 for *E. bieneusi*.

### 2.4. Statistical Analysis

The prevalence rates of *Cryptosporidium* spp., *G. duodenalis*, and *E. bieneusi* among different regions, types, genders, and age groups were calculated using SPSS v18.0 software (IBM, Armonk, NY, USA). Any difference was considered significant when the *p*-value was <0.05. The confidence intervals of proportions were calculated using the Wilson method.

## 3. Results

### 3.1. Prevalence of Cryptosporidium spp., G. duodenalis, and E. bieneusi in Cattle in Heilongjiang Province

Of the 1155 cattle fecal samples collected from 13 administrative regions of Heilongjiang Province, 64 were identified as *Cryptosporidium*-positive specimens using nested PCR amplification of the SSU rRNA gene (Table 2). The highest prevalence rate for *Cryptosporidium* spp. was recorded for Daqing County (11.9%, 24/201), followed by Mudanjiang County (7.8%, 11/141), and Hegang County (7.6%, 6/79). However, no specimens with *Cryptosporidium* infection were obtained from Jiamusi County, Qiqihar County, Shuangyashan County, or Suihua County. The prevalence rate for *Cryptosporidium* spp. differed significantly among the 13 regions (*p* < 0.001). Furthermore, the overall prevalence rates for *Cryptosporidium* spp. in cattle younger than 12 months, cattle aged from 12 to 18 months, and cattle older than 18 months were 12.8% (35/273), 2.9% (12/421), and 3.7% (17/461), respectively. The prevalence rate for *Cryptosporidium* spp. also differed significantly among age groups (*p* < 0.001). The prevalence rate for *Cryptosporidium* spp. in male cattle (7.2%, 31/433) was higher than that in females (4.6%, 33/722); however, this difference was not statistically significant (*p* = 0.063). Likewise, there was no significant difference (*p* = 0.506) in *Cryptosporidium* spp. prevalence between dairy cattle (6.0%, 35/585) and beef cattle (5.1%, 29/570).

A total of 44 *G. duodenalis*-positive specimens were detected utilizing the nested PCR amplification of the bg gene, representing a 3.8% infection rate (Table 3). The highest prevalence rate for *G. duodenalis* was recorded for Suihua County (10.8%, 8/74) and Qitaihe County (10.8%, 7/65), followed by Harbin County (6.0%, 8/134). No G. *duodenalis*-positive specimens were obtained from Jixi County, Jiamusi County, or Yichun County. A significant difference in the prevalence rate for *G. duodenalis* was observed among the 13 regions (*p* = 0.003). Specifically, the prevalence rate for *G. duodenalis* in beef cattle (5.3%, 30/570) was significantly higher than the rate in dairy cattle (2.4%, 14/585) (*p* = 0.011). Among age groups, the highest prevalence rate for *G. duodenalis* was found in cattle younger than 12 months (5.1%, 14/273), followed by cattle aged from 12 to 18 months (3.6%, 15/421), and cattle older than 18 months (3.3%, 15/461); however, this was not statistically significant (*p* = 0.416). In addition, the prevalence of *G. duodenalis* did not differ significantly between male cattle (4.4%, 19/433) and female cattle (3.5%, 25/722) (*p* = 0.426).

Regarding *E. bieneusi*, a total of 75 positive specimens were detected by utilizing the nested PCR amplification of the ITS gene, representing a 6.5% infection rate (Table 4). The highest prevalence rate for *E. bieneusi* was recorded for Qitaihe County (16.9%, 11/65), followed by Yichun County (13.3%, 8/60), and Daqing County (11.9%, 24/201). *E. bieneusi* was not found in cattle from Daxinganling County, Jiamusi County, or Qiqihar County. The prevalence rate for *E. bieneusi* differed significantly among the 13 regions (*p* < 0.001). Regarding age groups, we found that the overall prevalence rates for *E. bieneusi* infection in cattle younger than 12 months, cattle older than 18 months, and cattle aged from 12 to 18 months were 9.5% (26/273), 6.5% (30/461), and 4.5% (19/421), respectively. The prevalence rate differed significantly among age groups (*p* = 0.033). The prevalence rate for *E. bieneusi* in male cattle (7.2%, 31/433) was slightly higher than that in females (6.1%, 44/722), but the difference was not statistically significant (*p* = 0.477). Likewise, there was no significant difference between the prevalence rates for dairy cattle (5.5%, 32/585) and beef cattle (7.5%, 43/570) (*p* = 0.153).

Finally, only 1.1% (13/1155) of the fecal specimens were identified as being co-infected with two pathogens. Among these, four fecal samples were found to be positive for *Cryptosporidium* spp. and *G. duodenalis*, eight were positive for *Cryptosporidium* spp. and *E. bieneusi*, and one was positive for *G. duodenalis* and *E. bieneusi* (Table 5).

### 3.2. Molecular Characterization and Phylogenetic Analysis of Three Protozoan Pathogens in Cattle in Heilongjiang Province

In the present study, phylogenetic analysis was carried out for 64 *Cryptosporidium* spp.-positive sequenced specimens. Using the ML method, five known *Cryptosporidium* species were identified, including *C. andersoni* (3.3%, 38/1155), *C. bovis* (1.3%, 15/1155), *C. ryanae* (0.8%, 9/1155), *C. parvum* (0.1%, 1/1155), and *C. occultus* (0.1%, 1/1155) (Figure 2). Among these, *C. andersoni* was found in all areas where *Cryptosporidium* was detected, except Heihe County and Qitaihe County. *C. andersoni* was detected across different genders and age groups. At the same time, one *C. parvum*-positive sample was detected; this was obtained from a female dairy cow aged older than 18 months in Daqing County.

In addition, 44 *G. duodenalis*-positive samples obtained in the present study were identified as two assemblages, namely, assemblage A and assemblage E, by means of comparative analysis of bg gene sequences (Figure 3). Of these, the most common genotype was assemblage E (*n* = 43), which was distributed across different regions, types, genders, and age groups. Only one assemblage A-positive specimen was found; this was obtained from a female dairy cow aged older than 18 months in Qiqihar County.

A total of 75 *E. bieneusi*-positive sequenced specimens were identified by means of comparative analysis of the ITS gene; this revealed the existence of eight known genotypes, including the BEB4, BEB6, BEB8, J, I, CHS7, CHS8, and COS-I genotypes (Figure 4). Of these, the most commonly identified genotype was the J genotype (*n* = 33), followed by the BEB4 genotype (*n* = 13), and the I genotype (*n* = 11). Only one BEB8 genotype-positive specimen was found; this was obtained from a female beef cow aged older than 18 months in Qitaihe County. Phylogenetic analysis indicated that all the genotypes obtained in the present study belonged to group 2.

## 4. Discussion

Heilongjiang Province is one of the major agricultural provinces in China. It has long been known for its animal husbandry industry. The total amount of dairy cattle in Heilongjiang Province is more than 1.1 million, ranking it fourth among Chinese provinces, according to information released by the Department of Agriculture and Rural Affairs of Heilongjiang Province (http://nynct.hlj.gov.cn/nynct/c115443/202304/c00_31565039.shtml) accessed on 6 September 2023. Because healthy farm animals, including cattle, can be important animal reservoirs for enteric zoonotic pathogens, epidemiological surveys are one of the best ways to understand the molecular characteristics and diversity of enteric pathogens [29]. In the present study, for the first time, the prevalence rates for *Cryptosporidium* spp., *G. duodenalis*, and *E. bieneusi* in Heilongjiang Province were systematically and simultaneously determined.

*Cryptosporidium* spp. are important zoonotic intestinal protozoa that cause intestinal diseases in humans and animals [30]. In the present study, the overall infection rate of *Cryptosporidium* spp. In Heilongjiang Province among cattle fecal samples were found to be 5.5%, with prevalences of 6.0% and 5.1% in dairy and beef cattle, respectively. In comparison, previous research has shown that the infection rates of *Cryptosporidium* spp. in cattle from different provinces in China ranged from 0.7% to 29.9%. For dairy cattle, the *Cryptosporidium* spp. infection rate in the present study was higher than the corresponding rates previously reported for Ningxia Autonomous Region (1.6%, 23/1366) [31], Anhui Province (2.1%, 11/526) [32], Xinjiang Uyghur Autonomous Region (4.3%, 60/1391) [33], Shanxi Province (1.0%, 4/394) [7], Guangdong Province (4.4%, 63/1440) [34], and Gansu Province (5.1%, 58/1257) [35] but lower than those reported for Sichuan Province (14.4%, 40/278) [36], Yunnan Province (14.7%, 65/442) [37], Jiangxi Province (26.1%, 43/165) [38], Taiwan Province (26.5%, 60/226) [39], and Inner Mongolia (29.9%, 151/505) [40]. For beef cattle, the *Cryptosporidium* spp. infection rate in the present study was higher than that previously reported for the Tibet Autonomous Region (0.7%, 3/442) [41], Anhui Province (3.1%, 10/323) [32], and Shanxi Province (4.7%, 19/401) [10] but lower than that reported for Jiangxi Province (8.5, 24/283) [38]. Notably, the overall infection rate for *Cryptosporidium* spp. was found to be 19.15% (158/825) among 12–14-month-old yearlings of Heilongjiang cattle in a previous study that detected *Cryptosporidium* spp. oocysts by means of microscopy [21]. In the present study, using the nested PCR method, we found an infection rate of 12.8% for *Cryptosporidium* spp. in cattle younger than 12 months, a result in line with the previous finding.

Previous reports have also shown that *Cryptosporidium* spp. infection rates can be attributed to many factors, including sample sizes, climate conditions, management methods, and the immune status of animals. In one especially noteworthy study, the distribution of *Cryptosporidium* spp. in cattle was found to be related to the ages of hosts. It is reported that *C. parvum* infection was found predominantly in pre-weaned calves younger than 2 months, while *C. bovis* and *C. ryanae* infection generally occurred in post-weaned calves, and *C. andersoni* was much more prevalent in yearlings and adult cattle [42]. However, *C. parvum* has also been reported to be absent or present at a low rate on some cattle farms [43,44]. In the present study, *C. bovis* (*n* = 14) and *C. ryanae* (*n* = 7) were found to predominantly infect cattle younger than 12 months, in line with previously reported findings. However, we found only one instance of *C. parvum* in cattle aged older than 18 months, a finding inconsistent with previous reports on age-related infection. *C. parvum* is considered the most important zoonotic species, being responsible for the majority of human cryptosporidiosis outbreaks worldwide [45]. Moreover, domestic ruminants, particularly cattle, are regarded as the primary reservoir for *C. parvum* [46]. The above-mentioned cattle identified in the present study may, therefore, present a risk of zoonotic transmission, with potentially serious implications for public health. Furthermore, in the present study, apart from cattle younger than 12 months, *C. andersoni* was found to be present in all age groups and was clearly predominant, in line with previously reported results [47]. It is especially noteworthy that, in the present study, *C. occultus* (0.1%, 1/1155) was found in female beef cattle older than 18 months in Qitaihe County. This species has been previously reported in cattle, water buffalo, and domestic yaks; however, the primary hosts of this species have also been reported to be rodents [48]. Our result may have been due to cattle ingesting food contaminated with infected rodent feces; however, this protozoan resides only briefly in the intestines of cattle.

*Giardia duodenalis* has been identified as a prominent intestinal protozoan pathogen, with infections in cattle reported worldwide [49]. In the present study, the overall infection rate of *G. duodenalis* in fecal samples from Heilongjiang Province cattle was found to be 3.8%, with prevalences of 2.4% in dairy cattle and 5.3% in beef cattle. As was the case with *Cryptosporidium* spp., the infection rates of *G. duodenalis* for different Chinese provinces exhibited significant variation, with values between 1.0% and 41.2% being recorded. For dairy cattle, the rate of *G. duodenalis* infection found in the present study is similar to the reported rates for Ningxia Autonomous Region (2.1%, 29/1366) [31] and Guangdong Province (2.2%, 31/1440) [50] but higher than that reported for Gansu Province (1.0%, 14/1414) [51] and lower than that reported for Henan Province (7.2%, 128/1777) [52], Qinghai Province (10.0%, 39/389) [53], Xinjiang Uyghur Autonomous Region (13.4%, 69/514) [54], Jiangsu Province (20.6%, 281/1366) [55], Shaanxi Province (20.7%, 41/198) [56], Hubei Province (22.6%, 70/309) [57], Yunnan Province (27.5%, 144/524) [58], Inner Mongolia (29.5%, 149/505) [40], and Sichuan Province (41.2%, 126/306) [59]. For beef cattle, the *G. duodenalis* infection rate found in the present study is higher than that reported for Tibet Autonomous Region (3.8%, 17/442) [40] but lower than that reported for Inner Mongolia (9.2%, 10/108) [60], Shaanxi Province (16.76%, 29/173) [56], and Hubei Province (23.3%, 7/30) [57]. Notably, the overall prevalence of *G. duodenalis* was found to be 5.2% (42/814) in a previous study conducted in Heilongjiang Province, in which Lugol’s iodine staining was used [23]. In the present study, a similar infection rate (3.8%, 44/1155) was obtained using the nested PCR method. However, the previous data are now almost a decade old, and there is now a need to update the infection situation and evaluate any potential risks in Heilongjiang Province. Previous studies have indicated that the infection rate of *G. duodenalis* is related to animal age, with infection rates varying inversely with age [50,61]. However, in the present study, no significant differences were observed among the three age groups. Furthermore, the infection rate for beef cattle (5.3%, 30/570) in this study was significantly higher than that for dairy cattle (2.4%, 14/585). This is consistent with the findings of a study conducted in Belgium [62] but contrary to the infection rates reported for *G. duodenalis* in Scotland [63]. To date, eight assemblages (A–H) of *G. duodenalis* have been identified from different hosts, and three assemblages, namely, A, B, and E, have been isolated from cattle [64]. In the present study, assemblage E (3.7%, 43/1155) was found to be the predominant genotype among cattle in Heilongjiang Province, a result consistent with those of previous studies [65]. In recent years, some reports have indicated that assemblage E has zoonotic potential [66,67]. In light of this, it is noteworthy that one instance of assemblage A, the most common zoonotic genotype, was determined in cattle in the present study. This indicates that Heilongjiang Province cattle herders face the threat of zoonotic disease transmission.

In the present study, the infection rate for *E. bieneusi* in Heilongjiang Province cattle was found to be 6.5%, with prevalence rates of 5.5% in dairy cattle and 7.5% in beef cattle. The infection rates of *E. bieneusi* exhibit significant differences across different provinces in China. For dairy cattle, the *E. bieneusi* infection rate for Heilongjiang Province determined in the present study is higher than those reported for Yunnan Province (0.6%, 3/490) [68] and Anhui Province (3.0%, 16/526) [32] but lower than those reported for Jiangxi Province (7.9%, 13/165) [68], Jiangsu Province (13.0%, 177/1366) [55], Guangdong Province (15.7%, 61/388) [69], and Xinjiang Province (16.5%, 85/514) [70]. For beef cattle, the *E. bieneusi* infection rate for Heilongjiang Province determined in the present study is similar to that reported in Anhui Province (7.9%, 13/165) [32] but higher than those reported for Tibet Autonomous Region (2.5%, 11/442) [40] and Jiangxi Province (3.9%, 11/283) [71] and lower than that reported for Shanxi (22.4%, 90/401) [72]. Because the ITS gene has a high diversity rate, *E. bieneusi* can be categorized into many genotypes. This has been reported in cattle worldwide [73]. Phylogenetic analysis has indicated that all genotypes of *E. bieneusi* should be divided into 11 groups, with genotypes in groups 1 and 2 being associated with potential zoonotic or cross-species infections [74]. In the present study, eight known genotypes were detected from 75 *E. bieneusi*-positive samples based on the ITS gene locus in cattle, including CHS7 (2), BEB6 (5), COS-I (8), CHS8 (2), and BEB8 (1) belonging to group 1, and BEB4 (13), J (33), and I (11) belonging to group 2. Of these, J (44.0%, 33/75) was the dominant genotype. This phenomenon has also been reported in cattle from Xinjiang, Jiangsu, and Yunnan [55,68,71]. Furthermore, genotypes in group 2 were initially reported to be ruminant-adapted, and increasing numbers of zoonotic genotypes have also been found in this group, indicating an increasing risk of zoonotic infection between humans and animals. In the present study, all 75 positive samples in eight genotypes belonged to group 2, a result which indicated the presence of *E. bieneusi* infections, and consequent potential for zoonotic disease transmission, in Heilongjiang Province.

It is reported that these protozoans are widely distributed in the environment, including surface water sources [75]. The protozoan pathogens may be transmitted through water sources contaminated with the host’s feces and infect humans and animals, which increases the possibility of pathogen transmission. In addition, the present study has limitations that must be taken into consideration when interpreting the results obtained and the conclusions reached. The fecal samples transportation among cities in Heilongjiang Province might alter the quantity and quality of the DNA used for detection and molecular characterization purposes. Moreover, the lack of the genotyping data of the *C. parvum* isolates resulted in the knowledge of the implicated family subtypes. Thus, Future studies should overcome the limitations mentioned here and allow a better characterization of the diversity of these parasites.

## 5. Conclusions

To the best of our knowledge, this is the first study to systematically and simultaneously determine the prevalence and molecular characterization of *Cryptosporidium* spp., *G. duodenalis*, and *E. bieneusi* in cattle in Heilongjiang Province in Northeast China. Our findings indicate that a variety of species/genotypes are prevalent in cattle in Heilongjiang Province and that some of these are zoonotic, indicating a risk of zoonotic disease transmission in endemic areas. The findings reported in this paper not only expanded our understanding of the genetic composition and zoonotic potential of *Cryptosporidium* spp., *G. duodenalis*, and *E. bieneusi* in cattle but also provided basic data for the control and prevention of these intestinal protozoans in animals and humans.

## Figures and Tables

**Figure 1 animals-14-01635-f001:**
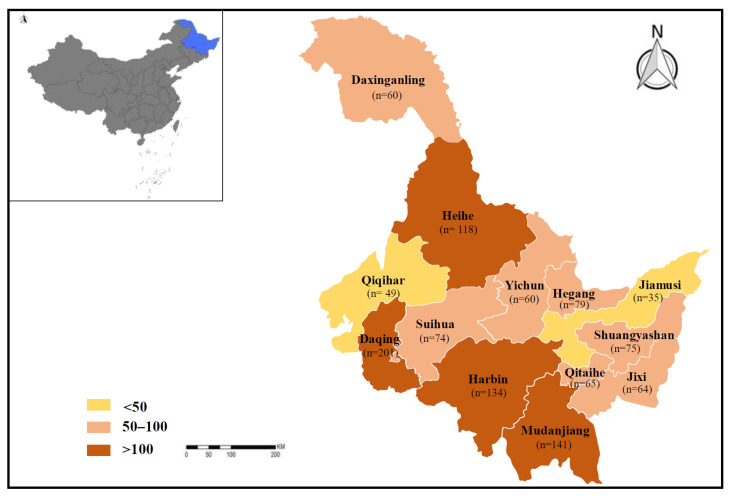
Geographical distribution of sampling sites of cattle in 13 regions of Heilongjiang Province in the present study.

**Figure 2 animals-14-01635-f002:**
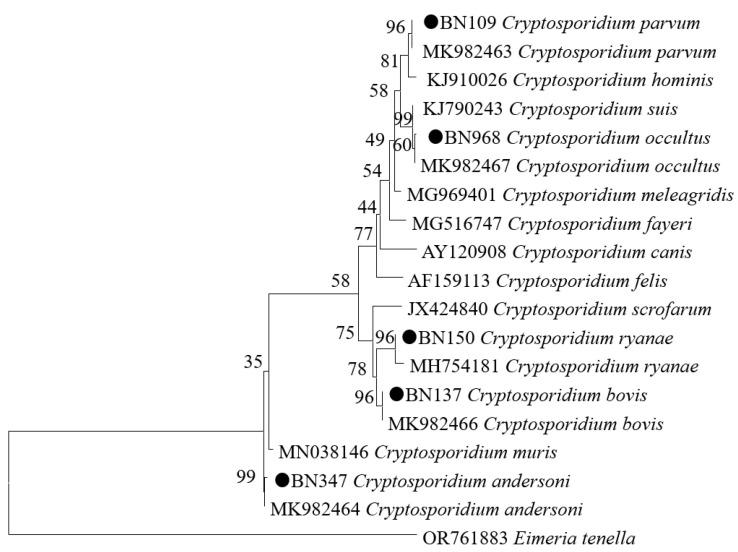
Phylogenetic relationships of *Cryptosporidium* spp. from cattle based on the *SSU rRNA* gene using maximum likelihood analysis. The circles before the bold sample names represent species identified in the present study. *Eimeria tenella* (AF026388.1) is used as the outgroup. Representative sequences of each sequence type in this study are included in the phylogenetic analysis.

**Figure 3 animals-14-01635-f003:**
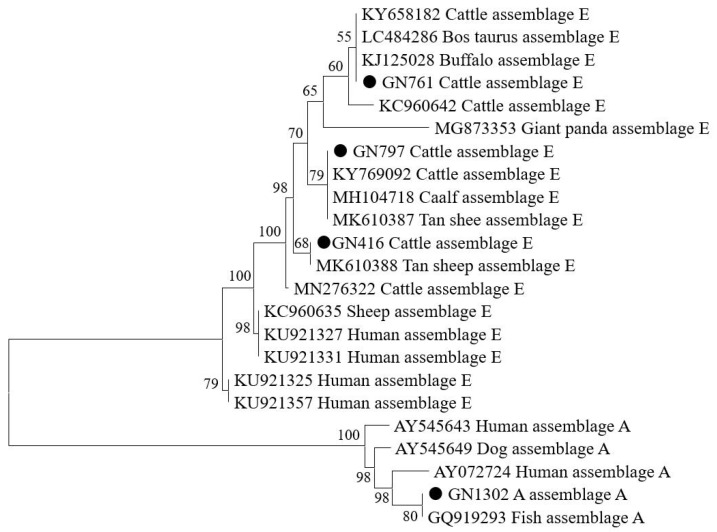
Phylogenetic relationships of *Giardia duodenalis* from cattle based on the *bg* gene obtained using maximum likelihood analysis. The circles before the bold sample names represent species identified in the present study. Representative sequences of each sequence type in this study are included in the phylogenetic analysis.

**Figure 4 animals-14-01635-f004:**
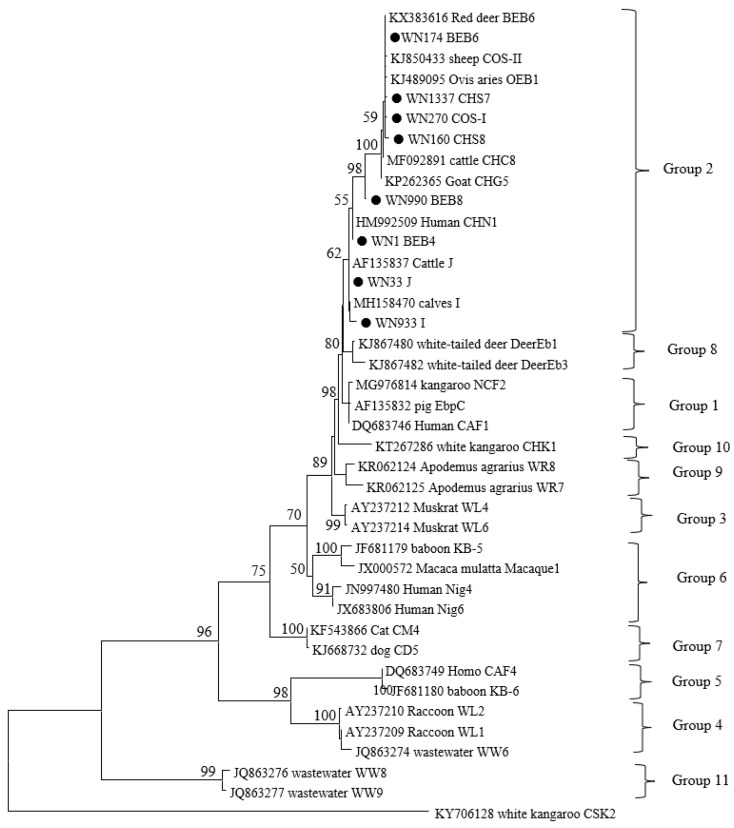
Phylogenetic relationships of representative sequences for the ITS genotypes of *E. bieneusi* identified in this study with reference sequences using maximum likelihood analysis. The circles before the bold sample names represent genotypes identified in the present study. Genotype CSK2 from white kangaroo (KY706128) is used as the outgroup.

**Table 1 animals-14-01635-t001:** Information of primer used for the molecular identification and/or the characterization of *Cryptosporidium* spp., *Giardia duodenalis*, and *Enterocytozoon bieneusi* in the present study.

Species	Loci	Primer ID	Primer Sequences (5′-3′)	Fragment Length (bp)	Temperature Annealing (°C)	Reference
*Cryptosporidium* spp.	*SSU rRNA* gene	CS-F1	TTCTAGAGCTAATACATGCG	830	55	[24]
		CS-R1	CCCATTTCCTTCGAAACAGGA			
		CS-F2	GGAAGGGTTGTATTTATTAGATAAAG			
		CS-R2	CTCATAAGG TGCTGAAGGAGTA			
*Giardia duodenalis*	*bg* gene	GD-F1	GAGGCCGCCCTGGATCTTCGAGACGAC	511	60	[25]
		GD-R1	GAACGAACGAGATCGAGGTCCG			
		GD-F2	CTCGACGAGCTTCGTGTT			
		GD-R2	TTCCGTRTYCAGTACAACTC			
*Enterocytozoon bieneusi*	ITS gene	EB-F1	GGTCATAGGGATGAAGAG	392	57	[26]
		EB-R1	TTCGAGTTCTTTCGCGCTC			
		EB-F2	GCTCTGAATATCTATGGCT			
		EB-R2	ATCGCCGACGGATCCAAGTG			

**Table 2 animals-14-01635-t002:** Occurrence of *Cryptosporidium* spp. in cattle in Heilongjiang Province.

Factor	Categories	No. Examined	No. of Positive	Prevalence% (95% CI)	OR (95% CI)	*p*-Value	Species (No.)
Region	Daqing	201	24	11.9% (7.5–16.4)	15.8 (2.1–118.2)	<0.001	*C. andersoni* (1), *C. bovis* (14), *C. parvum* (1), *C. ryanae* (8)
	Daxinganling	60	3	5.0 (0.0–10.50)	6.15 (0.6–60.5)		*C. andersoni* (3)
	Harbin	134	13	9.7 (4.7–14.7)	12.6 (1.6–97.6)		*C. andersoni* (12), *C. ryanae* (1)
	Hegang	79	6	7.6 (1.8–13.4)	9.0 (1.1–73.0)		*C. andersoni* (6)
	Heihe	118	1	0.9 (0.0–2.5)	1		*C. bovis* (1)
	Jixi	64	3	4.7 (0.0–9.9)	5.8 (0.6–56.5)		*C. andersoni* (3)
	Jiamusi	35	0	—	—		—
	Mudanjiang	141	11	7.8 (3.4–12.2)	9.9 (1.3–77.9)		*C. andersoni* (11)
	Qitaihe	65	1	1.5 (0.0–4.5)	1.8 (0.1–29.7)		*C. occultus* (1)
	Qiqihar	49	0	—	—		—
	Shuangyashan	75	0	—	—		—
	Suihua	74	0	—	—		—
	Yichun	60	2	3.3 (0.0–7.9)	4.0 (0.4–45.4)		*C. andersoni* (2)
Type	Dairy cattle	585	35	6.0 (4.1–7.9)	1.2 (0.7–2.0)	0.506	*C. andersoni* (14), *C. bovis* (14), *C. parvum* (1), *C. ryanae* (6)
	Beef cattle	570	29	5.1 (3.3–6.9)	1		*C. andersoni* (24), *C. bovis* (1), *C. occultus* (1), *C. ryanae* (3)
Gender	Female	722	33	4.6 (3.0–6.1)	1	0.063	*C. andersoni* (21), *C. bovis* (8), *C. occultus* (1), *C. parvum* (1), *C. ryanae* (2)
	Male	433	31	7.2 (4.7–9.6)	1.6 (1.0–2.7)		*C. andersoni* (17), *C. bovis* (7), *C. ryanae* (7)
Age	<12 Months	273	35	12.8 (8.9–16.8)	5.0 (2.6–9.8)	<0.001	*C. andersoni* (14), *C. bovis* (14), *C. ryanae* (7)
	12–18 Months	421	12	2.9 (1.3–4.4)	1		*C. andersoni* (9), *C. bovis* (1), *C. ryanae* (2)
	>18 Months	461	17	3.7 (2.0–5.4)	1.3 (0.6–2.8)		*C. andersoni* (15), *C. occultus* (1), *C. parvum* (1)
Total		1155	64	5.5 (4.2–6.9)			

Note: — indicates no data available.

**Table 3 animals-14-01635-t003:** Occurrence of *G. duodenalis* in cattle in Heilongjiang Province.

Factor	Categories	No. Examined	No. of Positive	Prevalence% (95% CI)	OR (95% CI)	*p*-Value	Assemblages (No.)
Region	Daqing	201	6	3.0 (0.6–5.3)	2.4 (0.3–20.3)	0.003	E (6)
	Daxinganling	60	2	3.3 (0.0–7.9)	2.7 (0.2–30.4)		E (2)
	Harbin	134	8	6.0 (2.0–10.0)	5.0 (0.6–40.4)		E (8)
	Hegang	79	1	1.3 (0.0–3.7)	1		E (1)
	Heihe	118	3	2.5 (0.0–5.4)	2.0 (0.2–19.9)		E (3)
	Jixi	64	0	—	—		—
	Jiamusi	35	0	—	—		—
	Mudanjiang	141	4	2.8 (0.1–5.6)	2.3 (0.3–20.7)		E (4)
	Qitaihe	65	7	10.8 (3.2–18.3)	9.4 (1.1–78.6)		E (7)
	Qiqihar	49	1	2.0 (0.0–6.0)	1.6 (0.1–26.6)		A (1)
	Shuangyashan	75	4	5.3 (0.2–10.4)	4.4 (0.5–40.2)		E (4)
	Suihua	74	8	10.8 (3.7–17.9)	9.5 (1.2–77.6)		E (8)
	Yichun	60	0	—	—		—
Type	Dairy cattle	585	14	2.4 (1.2–3.6)	1	0.011	E (13), A (1)
	Beef cattle	570	30	5.3 (3.4–7.1)	2.3 (1.2–4.3)		E (30)
Gender	Female	722	25	3.5 (2.1–4.8)	1	0.426	E (24), A (1)
	Male	433	19	4.4 (2.5–6.3)	1.3 (0.7–2.4)		E (19)
Age	<12 Months	273	14	5.1 (2.5–7.7)	1.6 (0.8–3.4)	0.416	E (14)
	12–18 Months	421	15	3.6 (1.8–5.3)	1.1 (0.5–2.3)		E (15)
	>18 Months	461	15	3.3 (1.6–4.9)	1		E (14), A (1)
Total		1155	44	3.8 (2.7–4.9)			

Note: — indicates no data available.

**Table 4 animals-14-01635-t004:** Occurrence of *E. bieneusi* in cattle in Heilongjiang Province.

Factor	Categories	No. Examined	No. of Positive	Prevalence% (95% CI)	OR (95% CI)	*p*-Value	Genotypes (No.)
Region	Daqing	201	24	11.9 (7.5–16.4)	9.4 (2.2–40.6)	<0.001	BEB4 (7), BEB6 (1), CHS8 (1), COS-I (3), I (1), J (11)
	Daxinganling	60	0	—	—		—
	Harbin	134	3	2.2 (0.0–4.7)	1.6 (0.3–9.7)		I (1), J (2)
	Hegang	79	4	5.1 (0.2–9.9)	3.7 (0.7–20.7)		I (3), J (1)
	Heihe	118	6	5.1 (1.1–9.0)	3.7 (0.7–18.8)		BEB6 (2), CHS8 (1),COS-I (3)
	Jixi	64	6	9.4 (2.2–16.5)	7.2 (1.4–36.7)		I (2), J (4)
	Jiamusi	35	0	—	—		—
	Mudanjiang	141	2	1.4 (0.0–3.4)	1		CHS7 (2)
	Qitaihe	65	11	16.9 (7.8–26.0)	14.2 (3.0–66.0)		BEB6 (1), BEB4 (1), BEB8 (1), J (8)
	Qiqihar	49	0	—	—		—
	Shuangyashan	75	7	9.3 (2.7–15.9)	7.2 (1.4–35.4)		J (4), I (2), BEB4 (1)
	Suihua	74	3	4.1 (0.0–8.5)	2.9 (0.5–18.0)		COS-I (2), BEB6 (1)
	Yichun	60	9	15.0 (6.0–24.0)	8.2 (1.7–39.0)		BEB4 (4), J (3), I (2)
Type	Dairy cattle	585	32	5.5 (3.6–7.3)	1	0.153	BEB4 (5), BEB6 (3), J (14), CHS7 (1), CHS8 (2), COS-I (6), I (1)
	Beef cattle	570	43	7.5 (5.4–9.7)	1.4 (0.9–2.3)		BEB4 (8), BEB6 (2), BEB8 (1), J (19), CHS7 (1), COS-I (2), I (10)
Gender	Female	722	44	6.1 (4.3–7.8)	1	0.477	BEB4 (7), BEB6 (3), BEB8 (1), J (15), CHS7 (1), CHS8 (2), COS-I (5), I (10)
	Male	433	31	7.2 (4.7–9.6)	1.2 (0.7–1.9)		BEB4 (6), BEB6 (2), J (18), CHS7 (1), COS-I (3), I (1)
Age	<12 Months	273	26	9.5 (6.0–13.0)	2.2 (1.2–4.1)	0.033	BEB4 (6), BEB6 (1), J (13), COS-I (3), I (3)
	12–18 Months	421	19	4.5 (2.5–6.5)	1		BEB4 (3), BEB6 (2), J (9), CHS7 (2), COS-I (1), I (2)
	>18 Months	461	30	6.5 (4.3–8.8)	1.5 (0.8–2.7)		BEB4 (4), BEB6 (2), BEB8 (1), J (11), CHS8 (2), COS-I (4), I (6)
Total		1155	75	6.5 (5.1–7.9)			

Note: — indicates no data available.

**Table 5 animals-14-01635-t005:** Occurrence of co-infection of three pathogens in cattle in Heilongjiang Province.

Region	Type		Gender		Age		
	Dairy Cattle	Beef Cattle	Female	Male	<12 Months	12–18 Months	>18 Months
Daqing	*C. ryanae* + E (2), *C. bovis* + J (2), *C. ryanae* + BEB4 (1)	*C. ryanae* + BEB4 (2)	*C. ryanae* + E (1), *C. bovis* + J (1)	*C. ryanae* + E (1), *C. ryanae* + BEB4 (3), *C. bovis* + J (1)	*C. ryanae* + E (2), *C. ryanae* + BEB4 (3), *C. bovis* + J (2)	—	—
Harbin	—	*C. andersoni* + E (2), *C. andersoni* + J (1), *C. andersoni* + I (1)	*C. andersoni* + E (2), *C. andersoni* + J (1), *C. andersoni* + I (1)	—	*C. andersoni* + E (1), *C. andersoni* + I (1)	—	*C. andersoni* + E (1), *C. andersoni* + J (1)
Qitaihe	—	*C. occultus* + BEB4 (1), E + J (1)	*C. occultus* + BEB4 (1)	E + J (1)	—	E + J (1)	*C. occultus* + BEB4 (1)

Note: — indicates no data available.

## Data Availability

The datasets supporting the results of this article have been submitted to GenBank and accession numbers are shown in the article.

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
