# Peer review of "Prevalence and Molecular Characterization of Cryptosporidium spp., Giardia duodenalis, and Enterocytozoon bieneusi in Cattle in Heilongjiang Province, Northeast China"

_animals, 2024, doi:10.3390/ani14111635_

Round 1

Reviewer 1 Report

Comments and Suggestions for Authors

The study by Gao et al. is well-written and conducted, with an important contribution to the knowledge of the molecular epidemiology of three zoonotic pathogens in cattle. I support its further processing after appropriate minor modifications as outlined below:

L25: when you present overall prevalence values, please indicate in brackets the values of 95% confidence interval.

L51: please ensure appropriate citations after the end of each statement/sentence/paragraph.

L56: “worldwide [8].” – the reference no. 12 would be more suitable for citation

L106: “we sought to conduct” – please avoid the use of personal mode verb formulations, may sound unprofessional. Revise this concern throughout the manuscript.

L107: it would be useful, also, to mention that the study aimed to evaluate the public health potential of the isolates

Overall, the title, abstract, and introduction are well-written and effectively convey the context and importance of the study.

L111: the total number of processed samples is unclear. It would be useful to know if the authors took into consideration a sample size calculation based on a previous prevalence

L119: “until use.” – this can be from a few hours to several days, months, or even years. Please be more specific.

L135-137: please uniformly mention for all of the used reagents the production company name, city, and country

L139: “Primer” – lowercase

Within the results section, the authors must refer/mention the identified species distribution of the screened region. In Table 2 the presentation of the mentioned species is unclear. The same request for Giardia spp. and Enterocytozoon bieneusi.

L229: which was the selection strategy for the used positive sequenced specimens in the phylogenetic three construction? The same question for Giardia and Enterocytozoon?

Overall, the study demonstrated a widespread occurrence of Cryptosporidium and Giardia infections in the investigated region. The authors must refer to possible infection sources like contaminated surface waters, as has been previously demonstrated in other investigations (e.g. doi: 10.1089/vbz.2017.2155). The authors can consult and cite the indicated reference.

At the end of the discussion section, the authors must refer to the study's limitations. In this regard, they must mention the lack of the subtyping of the C. parvum isolates, resulting in the knowledge of the implicated family subtypes. This can constitute further research directions!

In my opinion, the paper's novelty and the way it presents the properties under investigation make a strong case for its further processing after appropriate modifications as highlighted in my review.

Reviewer 2 Report

Comments and Suggestions for Authors

Dear authors,

In this study, the prevalence and distribution of Cryptosporidium spp., Giardia duodenalis, and Enterocytozoon bieneusi, which are important enteric pathogens of cattle, were investigated by molecular methods in Heilongjiang province of Northwest China and current epidemiological data on the status of the pathogens in cattle herds were obtained. Moreover, DNA sequence analysis of the positive samples detected within the scope of the study was performed and the genotypes circulating in the region were revealed. Although the data obtained within the scope of the study are important, it is evaluated that corrections should be made within the scope of the recommendations listed below.

best regards

Line 6 The name of the last author is not written, only the symbol "*". (the last word of authors names).

Abstract

In the summary section of the manuscript, the authors provide information about the number of samples used in the study, the pathogens identified, their prevalence, and phylogenetic characteristics. They also interpreted the results obtained and presented information about the contribution of the study to science in this section. However, the number of words in the abstract section is more than the specified limit in the Instructions for Authors of the journal.

Introduction

In the introduction section of the manuscript, one paragraph information about the Cryptosporidium spp., Giardia duodenalis, and Enterocytozoon bieneusi species, genotypes, and host species investigated in the study is presented. Since the study was conducted on cattle, more information about the effects of pathogens on cattle health should be added in this part.  However, it is seen that the effects of pathogens on humans are given more space in this section. In addition, it is suggested that the following corrections be made regarding the introduction.

Lines 44-48. This part should be rewritten. The sentence is too long and therefore this is not reader-friendly. In addition, there is no connection seen between the first and last sentences of this section.

Line 49-63. Repetitive sentences (such as post-weaned, pre-weaned, and most common) in this paragraph are used and these parts should be removed.

Line 77 Please remove the dot before the reference.

Line 78 Please do not start the paragraph with an abbreviation. Please write Enterocytozoon bieneusi instead of E. bieneusi.

Material and Methods

In the material-methods section of the manuscript, the authors gave detailed information about the regions sampled within the scope of the study. They also described in detail the isolation of genomic DNA from fecal samples and the PCR process performed for the detection of Cryptosporidium spp., Giardia duodenalis, and Enterocytozoon bieneusi from these DNAs. In addition, phylogenetic analysis of the pathogens identified in the study is also described in this section. Only one correction is suggested below. 

Line 140 Please add comma before “and”

Line 145 The PCR annealing temperatures in the table-1 are different from the annealing temperatures given in this line. This difference should be corrected.

Results

In this part of the study, the distribution of the prevalence of Cryptosporidium spp, Giardia duodenalis, and Enterocytozoon bieneusi in fecal samples according to sampling areas, sex and age groups was presented. In this section, information about the genotypes of Cryptosporidium species, Giardia duodenalis, and Enterocytozoon bieneusi detected in the study area were also presented. However, some corrections are suggested in this section. Especially Table 4 and table 5 should be reviewed carefully. 

Line 193. Please replace duodenalis-positive with G. duodenalis-positive.

In Table 4, when the number of Enterocytozoon bieneusi detected according to the regions sampled is summed, it is 74, but in other parts of the table, this number is 75. This difference should be corrected. 

Again in Table 4 although Enterocytozoon bieneusi was not detected in Qiqihar city, it is understood that the pathogen was detected in 7 samples and these were genotyped. There are probably shifts in the table rows and these shifts should be corrected.

Line 208 Please remove “infection”, because in this study pathogens were researched and detected. Any data is present about clinical symptoms in animals.

In Table 5, a genotype called “E7” is mentioned in the list of mixed infections in the Qitaihe region, but genotype “E7” is not mentioned anywhere in the manuscript. This section should be corrected.

Discussion

In the discussion section of the manuscript, the researchers discuss the results of the study in comparison with previous studies. In addition, the contribution of the results obtained within the objectives of the study to science is also given in this section. Some corrections are suggested below. 

Line 291, 294, 342 Please add a comma before “and”

Line 326 Please do not start the paragraph with an abbreviation. Please write Giardia duodenalis instead of G. duodenalis.

Line 369 Please remove extra space before 13/165, [68], Jiangsu, (13.0%.., 177, [55], and Guangdong  

Reviewer 3 Report

Comments and Suggestions for Authors

This manuscript is well organized and written. Only a few minor changes are requested.

1. L8: 'Rural Affair'

2. L13: '... and molecularly detect the prevalence of three ...'

3. L16: '.... and that some of them are zoonotic, ...'

4. L22: '... intestinal protozoans, the ...'

The study and manuscript are very fine.
